# The Role of the Gut Microbiome in Psychiatric Disorders

**DOI:** 10.3390/microorganisms10122436

**Published:** 2022-12-09

**Authors:** Ioana-Maria Andrioaie, Alexandru Duhaniuc, Eduard Vasile Nastase, Luminița Smaranda Iancu, Cătălina Luncă, Felicia Trofin, Dana-Teodora Anton-Păduraru, Olivia-Simona Dorneanu

**Affiliations:** 1Microbiology Department, University of Medicine and Pharmacy “Grigore T. Popa”, 700115 Iasi, Romania; 2Clinical Hospital of Infectious Diseases “Sf. Parascheva”, 700116 Iasi, Romania; 3Infectious Diseases Department, University of Medicine and Pharmacy “Grigore T. Popa”, 700115 Iasi, Romania; 4National Institute of Public Health, Iasi Regional Center for Public Health, 700465 Iasi, Romania; 5Emergency Clinical Hospital for Children “Sf. Maria”, 700309 Iasi, Romania; 6Pediatrics Department, University of Medicine and Pharmacy “Grigore T. Popa”, 700115 Iasi, Romania

**Keywords:** gut microbiome, microbiota–gut–brain axis, psychiatric disorders, depression, anxiety, autism, schizophrenia, probiotics

## Abstract

The role of the gut microbiome in mental health has been of great interest in the past years, with several breakthroughs happening in the last decade. Its implications in several psychiatric disorders, namely anxiety, depression, autism and schizophrenia, are highlighted. In this review were included relevant studies on rodents, as well as human studies. There seems to be a connection between the gut microbiome and these pathologies, the link being emphasized both in rodents and humans. The results obtained in murine models align with the results acquired from patients; however, fewer studies regarding anxiety were conducted on humans. The process of sequencing and analyzing the microbiome has been conducted in humans for several other pathologies mentioned above. Additionally, the possible beneficial role of probiotics and postbiotics administered as an aid to the psychiatric medication was analyzed.

## 1. Introduction

The gut microbiome is a complex community of different microorganisms that live together in a symbiotic state within the human host. The most abundant species found within the gut belong to *Firmicutes* phyla, especially *Clostridium* and *Lactobacillus* species, and *Bacteroidetes* phyla, especially *Bacteroides* and *Prevotella* species [1,2]. Other less encountered species belong to other phyla such as *Proteobacteria*, *Actinobacteria*, *Fusobacteria*, *Spirochaetes*, *Verrucomicrobia* and *Lentisphaerae* [1,2]. The gut microbiome plays an important role in several functions such as digestion, producing essential metabolites and the development of the immune system [3]. Dysbiosis of the gut microbiome is correlated with various diseases such as colorectal cancer [4,5], inflammatory bowel disease (IBS) [6,7], diabetes and metabolic syndrome [3]. Another important role of the gut microbiome is its involvement in the immune modulation and the development of the nervous system, as several studies have shown a connection between dysbiosis of the gut microbiome and neuropsychiatric disorders [8]. The microorganisms of the gut microbiome are communicating with the brain through the gut–brain axis [9]. A better understanding of the function of the microbiome has expanded the concept of the gut–brain axis to the microbiota–gut–brain axis, accentuating the importance of the microbiome in the gut–brain communication [10]. The microbiota–gut–brain axis is controlled by several systems such as the central nervous system, the enteric nervous system and the autonomous nervous system and by neuroendocrine and neuroimmune pathways. The neuronal function can be altered directly or indirectly by the microbiota through the production of vitamins and neuroactive microbial metabolites such as short-chain fatty acids (SCFA) [11]. Alterations in the gut microbiome can also regulate the synthesis of several central neurotransmitters such as melatonin, gamma-aminobutyric acid (GABA), serotonin, histamine and acetylcholine [12]. Studies on germ-free mice have shown that the gut microbiome is necessary for the normal development of the brain, particularly for the development of hippocampal and microglial morphology [13,14]. Additional research stated that GF mice present more microglia in the entire brain when compared with controls. The microglia also show abnormal morphology, with longer more complex processes [13]. A study on the maturation and function of microglial cells in germ-free mice has shown that the absence of microbiota or a limited diversity of microbiota is correlated with microglial impairment, and recolonization with a complex microbiota partially restores features of the microglial cells [15,16]. Moreover, germ-free mice that lack all gut bacteria develop specific impairments of immune, neuronal and metabolic functions, as well as gastrointestinal tract abnormalities [13]. Sudo et al., (2004) have shown that exposure to native microbiota in a stage of early development, when brain plasticity is still conserved, is necessary for the hypothalamic–pituitary–adrenal (HPA) system to become ready for inhibitory neural regulation [17]. Furthermore, germ-free (GF) mice displayed a hyperresponsive HPA axis after stress when compared with specific-pathogen-free (SPF) mice [18]. Administration of oral antimicrobials to SPF mice resulted in a change in the composition of the microbiota that was accompanied by an increase in exploratory activity [19].

Therefore, this paper aims to analyze the role of the microbiome and the gut–brain axis in the development of certain psychiatric disorders, as well as the possible beneficial role of probiotics administered as an aid to the psychiatric medication.

The methods of this review article were based on the use of the PubMed database to search for all related published studies. The selection was based on the keywords “gut microbiome”, “gut–brain axis”, “psychiatric disorders”, “depression”, “anxiety”, “autism”, “schizophrenia”, “probiotics” and “postbiotics”. We found 2041 articles and, after the exclusion process, 57 qualifying studies remained. The inclusion criteria were: (1) the year of publication (2000–2022); (2) the relevance of the title and of the abstract; (3) the availability of the full-text article. The exclusion criteria were: (1) studies also related to neurological diseases not only psychiatric ones; (2) title and abstract irrelevant to the aim of our study (Figure 1).

## 2. The Intestinal Microbiome in Psychiatric Pathology

### 2.1. The Role of the Gut Microbiome in Anxiety

Anxiety disorder is one of the most frequent psychiatric conditions, which has a negative impact on academic and professional performance and social and family interactions [20]. Anxiety disorders are defined by a diverse mental health symptomatology, including hyper-arousal and excessive fear and worry, usually associated with behavioral and cognitive distress responses [21].

Multiple studies have shown a comparison between the behavior of GF mice and that of SPF mice in a series of tests: the open-field box test [22], the light–dark box test [22] and the elevated plus maze test (EPM) [18,22,23]. The tests are based on the fact that rodents display aversion towards large, brightly lit, open and unknown environments and elevated areas [24,25]. A first set of experiments had both the GF mice and the SPF mice placed in an open-field box, where their exploratory activity was measured for 60 min. GF mice traveled a greater distance and explored the area significantly, including the center of the box. GF mice have also shown higher levels of rearing in comparison with SPF mice [22]. Regarding the light–dark box test, GF mice were observed to be spending more time in the light compartment compared with SPF mice [22]. In the EPM test, GF mice were observed to be spending more time in the open arm than the SPF mice. GF mice also visited the ends of the open arms more times in comparison with SPF mice [22]. GF mice were also observed to be spending less time in the closed arms compared with SPF mice [18]. These studies show that GF mice seem to be less anxious than SPF mice. Another study concluded that GF mice showed a lower anxiety-like behavior in the EPM test [26].

In a recently published scientific report, the authors divided the mice into two different categories, high anxiety (HA) and low anxiety (LA), and analyzed the composition of each group’s gut microbiome. In HA mice they observed an increased abundance of bacteria included in the genera *Ruminiclostridium*, *Ruminococcaceae_UCG-014* and *Oscillibacter* from the *Ruminococcaceae* family, as well as *Clostridiaceae* and *Clostridiales*, while microorganisms included in the genus *Bacteroides* from the *Bacteroidaceae* family were in a lower abundance [27]. Kang et al. (2014) concluded previously that the abundance of *Ruminococcaceae* had correlated negatively with the percent of time spent by the mice in light, therefore lower *Ruminococcaceae* levels correlated with less anxiety [28].

In a study in which mice were exposed to stressors for an extended period of time, it was observed that the presence of the genus *Alistipes* was highly increased [29]. Another study on mice showed that infection with the pathogenic bacteria called *Citrobacter rodentium* induced an anxiety-like behavior [30].

Transplantation of fecal microbiota between mice with different levels of anxiety showed that the microbiota can alter brain chemistry and change the behavior of mammals [31].

The fecal microbiota of patients diagnosed with generalized anxiety disorder (GAD) was analyzed by Jiang et al., (2018). Eight genera exhibited a lower prevalence compared with the healthy controls (HCs): *Faecalibacterium*, *Eubacterium*, *Lachnospira*, *Butyricicoccus*, *Sutterella*, *Bacteroidetes*, *Ruminococcus gnavus* and *Fusobacterium* [32]. There is a consensus regarding the studies performed in mice, however, in humans, in opposition to mice, a lower prevalence of *Ruminococcus* has been observed (Table 1).

### 2.2. The Role of the Gut Microbiome in Depression

Major depressive disorder (MDD) is a common mental disorder affecting up to 15% of the general population [33]. Depressed patients are characterized by a diminished quality of life accompanied by mood symptoms [34]. Depression has been correlated with an abnormal function of the HPA axis, and a link between the gut microbiome and HPA axis has been described. Exaggerated corticosterone and adrenocorticotrophin levels were observed in GF mice in response to stress compared with SPF mice. Moreover, the microbiota could have an impact on the central nervous system by activating the stress circuits at the neuronal level. Oral administration of pathogenic bacteria such as *Citrobacter rodentium* and *Campylobacter jejuni* showed that gut microbes, including food pathogens for humans such as *C. jejuni*, can activate vagal pathways that produce a stress response [35].

An important factor that could alter the gut microbiome, therefore having a role in the development of depression, is diet. In a large cross-sectional study, the severity of depression was inversely correlated with folate and vitamin B12 intake [36]. Moreover, in a 12-week study suitably named the SMILES trial, patients with MDD were advised to make a dietary change and to go on a modified Mediterranean diet (MD). The results have persuasively shown that diet can attenuate MDD symptoms [37]. The main food groups in the Mediterranean diet, namely vegetables, fruits and pulses, are linked to high amounts of bacteria that have anti-inflammatory properties. In individuals with a higher MD adherence, an increased presence of *Bacteroidetes* and a lower *Firmicutes–Bacteroidetes* ratio was linked to a lower animal protein intake [38].

It has been shown that after transferring the gut microbiome of a depressed patient into a healthy rodent, the latter displayed depressive symptoms due to disturbances in the carbohydrate and amino acid metabolism. This suggests there is a link between the gut microbiome and the pathophysiology of depression [35,39,40].

A study that included 34 depressed patients and 33 healthy subjects with corresponding gender, age and ethnicity was conducted to assess the involvement of the gut microbiome in depression. Their extracted fecal DNA was sequenced. A fecal microbiota transplantation (FMT) to rats was conducted. The fecal samples for the transplant were obtained from the three most severely depressed male patients as well as three HCs with corresponding age and sex. Regarding the depressed patients, it has been observed that, at the family level, the proportions of *Prevotellaceae* were low, while those of *Thermoanaerobacteriaceae* were high. At the genus level, the proportions of *Eggerthella*, *Holdemania*, *Gelria*, *Turicibacter*, *Paraprevotella* and *Anaerofilum* were increased, while those of *Prevotella* and *Dialister* were decreased. The rats who received FMT from depressed patients have shown anxiety-like behavior, as demonstrated by fewer visits to the open arms in the EPM and a lower amount of time spent in the center of the open field. There were no alterations in overall activity, as shown in the total visits to the closed and open arms in the EPM and total activity in the open field [40]. Several studies assessed the potential role of the FMT on patients with anxiety and depression and associated inflammatory bowel disease (IBS) refractory to traditional therapies. Several scores were used to assess psychiatric symptoms, such as hospital anxiety and depression (HAD) tests, Hamilton rating scale for depression (HAM-D) and Hamilton rating scale for anxiety (HAM-A). In all studies, an amelioration in IBS-related symptoms was observed, as well as a temporary improvement in HAD, HAM-D and HAM-A scores. However, fewer studies were conducted on humans, so FMT as a potential beneficial therapy needs to be further explored [41].

Another study noticed that in MDD patients there is an abundance of *Actinobacteria*, while *Bacteroidetes* are decreased [39]. A study that used the quality of life (QoL) questionnaire reported that *Faecalibacterium* and *Coprococcus (*butyrate producing bacteria) were positively associated with QoL scores and were found to be reduced in depression. The role of the butyrate, which is a short-chain fatty acid, is to strengthen the epithelial protective barrier and to reduce intestinal inflammation. On the other hand, *Flavonifractor* has been observed to be increased in depressive patients [42].

Another study analyzed 46 patients with depression (29 active-MDD and 17 responded-MDD), as well as 30 HCs. Active-MDD patients are those with acute MDD who had a high score on a depression scale, which indicates a clinically significant depression, while responded-MDD patients are characterized by a 50% reduction in depression scale scores in response to 4 weeks of depression treatment. A statistically significant difference in the gut microbiome between the groups was observed. The active-MDD group showed significantly higher proportions of *Acidaminococcaceae*, *Enterobacteriaceae*, *Fusobacteriaceae*, *Porphyromonadaceae* and *Rikenellaceae* compared with the HCs, while the levels of *Bacteroidaceae*, *Erysipelotrichaceae*, *Lachnospiraceae*, *Prevotellaceae*, *Ruminococcaceae* and *Veillonellaceae* were lower compared with the HCs. At the genus level, they observed differences between 72 genera (15 predominant genera and 57 less-predominant genera). *Alistipes*, *Blautia*, *Clostridium* XIX, *Lachnospiracea incertae sedis*, *Megamonas*, *Parabacteroides*, *Parasutterella*, *Phascolarctobacterium*, *Oscillibacter* and *Roseburia* were relatively more abundant in the active-MDD group compared with the HC group, while *Bacteroides*, *Dialister*, *Faecalibacterium*, *Prevotella* and *Ruminococcus* were relatively more abundant in the HC group compared with the active-MDD group. Regarding the responded-MDD group, a higher abundance of *Acidaminococcaceae*, *Bacteroidaceae*, *Enterobacteriaceae*, *Porphyromonadaceae* and *Rikenellaceae* has been noticed compared with the HCs, while the presence of *Lachnospiraceae*, *Ruminococcaceae* and *Veillonellaceae* was decreased compared with the HCs. At the genus level, there were several significant differences between the responded-MDD and the HC groups. *Alistipes*, *Bacteroides*, *Parabacteroides*, *Phascolarctobacterium* and *Roseburia* were increased, while *Escherichia/Shigella*, *Oscillibacter*, *Dialister*, *Faecalibacterium*, *Prevotella* and *Ruminococcus* were significantly decreased in the responded-MDD group compared with the HC group. The study concluded that the fecal microbial diversity of active-MDD was unexpectedly higher, although the exact repercussion of high fecal bacterial diversity remains unclear [43]. Naseribafrouei et al., (2014) showed increased levels of *Bacteroidales* at the order level, *Oscillibacter* and *Alistipes* at the genus level and decreased levels of *Lachnospiraceae* in MDD patients compared with HCs [44]. Valles-Colomer et al. (2019) showed increased levels of *Bacteroides* enterotype 2 and decreased levels of *Coprococcus* and *Dialister* in MDD patients compared with HCs [42].

Eight bacterial taxa were identified to be the biomarkers of depression in children: *Prevotella*, *Bifidobacterium*, *Escherichia/Shigella*, *Agathobacter*, *Gemmiger*, *Streptococcus*, *Collinsella* and *Klebsiella.* MDD in children is considered to be best predicted by *Streptococcus* [45]. The studies seem to be in agreement on the prevalence of several microorganisms in depression; however, there is a difference in the presence of *Bacteroidales* between the data reported (Table 2).

### 2.3. The Role of the Gut Microbiome in Autism

Autism spectrum disorder (ASD) is a complex disorder characterized by deficiencies of language and social interaction skills and repetitive behavioral patterns, as well as immune dysregulation and gastrointestinal manifestations [46,47,48,49].

The possible implication of the gut microbiome in ASD was first put into question when several reports of children developing regressive ASD after antibiotic treatment for chronic otitis media arose [50,51]. Wimberley et al. (2018) stated that otitis media is more frequently related to autism than other infections. However, broad spectrum antibiotic usage could not be linked to the association between otitis media and autism [52]. A study that included 13 ASD children and 8 HCs noted that the intestinal flora of ASD children included higher *Clostridium* and *Ruminococcus* species. Furthermore, some of those species were only present in children with autism and such species could be neurotoxin producers, which may contribute to the ASD impairment. [53]. A clinical trial regarding the effect of vancomycin in 11 children with regressive ASD was conducted. Each subject was administered vancomycin for 8 weeks, followed by oral probiotics (*Lactobacillus acidophilus*, *Lactobacillus bulgaricus* and *Bifidobacterium bifidum*) for 4 weeks. Eight of the eleven children showed notable behavioral improvements. However, most of the children displayed significant behavioral deterioration after vancomycin cessation [54]. Despite the significant limitations of this study (small treatment group, lack of an untreated control group), it has shown an important first supposition regarding the implications of the gut microbiome in ASD [49].

A population-based cohort study mentioned a link between the maternal use of antibiotic therapy during pregnancy and the later development of autistic symptomatology in the child [55].

A study in which 33 autistic subjects, 7 non-autistic siblings and 8 control subjects were included has shown that there is a significantly higher diversity found in the gut microbiome of autistic subjects compared with controls, the most common ones being *Bacteroidetes* and *Firmicutes*. The high diversity in the intestinal microbiome of autistic children might be formed by harmful genera that might contribute to the intensity of autistic manifestations. Several bacteria produce short-chain fatty acids such as butyric, propionic, acetic and valeric acids as part of their metabolism [45,46]. *Firmicutes* families such as *Ruminococcaceae*, *Lachnospiraceae*, *Erysipelotrichaceae* and *Clostridiaceae* represent the main producers of butyrate, while species of *Bacteroidetes*, namely *Bacteroides* spp., *Veillonella* spp. or *Blautia* spp. are the main producers of propionate. *Akkermansia municiphila* produces both propionate and acetate and *Bifidobacterium* spp. produces both acetate and lactate [45]. SCFAs have been shown to produce autistic symptoms in rats [47].

Dan et al. (2020) reported a higher abundance of *Firmicutes*, *Proteobacteria* and *Actinobacteria*, as well as increased levels of *Dialister*, *Escherichia-Shigella* and *Bifidobacterium*, while the *Bacteroidetes* proportion was decreased [56]. Zhai et al. (2019) discovered a higher proportion of *Bacteroides*, *Parabacteroides*, *Sutterella*, *Lachnospira*, *Bacillus*, *Bilophila*, *Lactococcus*, *Lachnobacterium* and *Oscillospira* in ASD children compared with HCs [57]. In their study, Zou et al. (2020) described an increased amount of *Bacteroidetes* and decreased amounts of *Firmicutes*, *Proteobacteria* and *Verrucomicrobia* at the phylum level, while at the genus level they reported a higher level of *Bacteroides*, *Prevotella*, *Lachnospiracea_incertae_sedis* and *Megamonas* and a lower level of *Clostridium XlVa*, *Eisenbergiella*, *Clostridium IV*, *Flavonifractor*, *Escherichia/Shigella*, *Haemophilus*, *Akkermansia* and *Dialister* [58]. Finegold et al. (2010) found an increase in the *Desulfovibrio* species, as well as *Bacteroidetes vulgatus,* in the fecal microbiota of children with severe ASD compared with HCs [46]. Ahmed et al. (2020) reported a higher abundance of *Bacteroides* and *Ruminococcus* in ASD children and their siblings, but not in the HCs [59]. As Ding et al. (2020) reported, there was a significant increase in unidentified *Lachnospiraceae*, *Clostridiales*, *Erysipelotrichaceae*, *Dorea*, *Collinsella* and *Lachnoclostridium* in ASD children, while *Bacteroides*, *Faecalibacterium*, *Parasutterella* and *Paraprevotella* were decreased [60]. Kang et al. (2013) showed increased levels of *Akkermansia* and decreased levels of *Prevotella*, *Coprococcus* and *Veillonellaceae* in children with ASD compared with HCs [61], while Wang et al. (2013) showed increased levels of *Sutterella* and *Ruminococcus torques* in ASD children compared with HCs [62]. These studies show contradictory results regarding the presence of *Bacteroides* and *Prevotella* in ASD (Table 3).

### 2.4. The Role of the Gut Microbiome in Schizophrenia

Schizophrenia (SCZ) is a destructive illness associated with hallucinations, delusions and thought disorders affecting up to 1% of the general population worldwide [63,64].

A study that included a total of 63 patients with SCZ and 69 HCs stated that in the case of patients with SCZ there was a lower within-sample diversity regarding the microbial composition of fecal specimens. There have been identified by sequencing 77 differential operational taxonomic units (OTUs), which discriminate between the two groups. Twenty-three OTUs were increased in patients with SCZ in comparison with HC, these being *Veillonellaceae*, *Prevotellaceae*, *Bacteroidaceae* and *Coriobacteriaceae*, while fifty-four OTUs were decreased, namely *Lachnospiraceae*, *Ruminococcaceae* and *Enterobacteriaceae*. Afterwards, a gut microbiome transplantation from patients with SCZ to mice was conducted. As a result of the FMT, mice started to exhibit SCZ-like manifestations, such as hyperactivity measured by a greater total distance traveled in the open field test, as well as lower anxiety, determined by a higher time spent in the exposed center area. Similar results were obtained in the forced swimming test: the mice were placed one by one in plexiglass cylinders that were filled with 15 cm of water and were considered immobile if they stayed floating in a vertical position, keeping only their head above the water. The duration of immobility was notably decreased in the SCZ FMT mice compared with the HC mice, indicating lower depressive-like behavior [64]. Another study indicated that prenatal microbial infection was shown to increase the frequency of SCZ and SCZ spectrum disorders by 10–20 times in the newborn [65]. Disruption of the microbiota–gut–brain axis might be involved in the development of schizophrenia, but further research in this area is needed [64,65,66].

## 3. The Effects of Probiotics in Psychiatric Disorders

Probiotics are live microorganisms with beneficial effects on human health and they frequently include lactic-acid-producing bacteria from the *Bifidobacterium* and *Lactobacillus* genera. Probiotics that alter cognitive functions, mood and anxiety are referred as psychobiotics and, recently, this term was expanded to all interventions that alter the microbiota and influence the relationships between bacteria and the brain [67]. A meta-analysis showed that supplementation with probiotics in healthy individuals led to a significant improvement in preclinical psychological symptoms of anxiety and depression [68]. The mechanism through which probiotics induce beneficial effects on psychiatric disorders is not well understood, but it is thought that probiotics promote biosynthesis of neurotransmitters such as GABA, dopamine, serotonin, norepinephrine and acetylcholine, which improve mood symptoms [69]. The role of probiotics in maintaining intestinal and brain health was demonstrated in a study on Rag1 knockout mice who lack mature B and T-cells and therefore have abnormal gut and neuronal functions. In this case, the probiotic treatment partially normalized these impairments [70].

### 3.1. The Effects of Probiotics in Depression

Akkasheh et al. (2016) conducted a study on 40 patients diagnosed with MDD who were randomly allocated into two groups: one group received probiotic supplements (*n* = 20) and one group received placebo (*n* = 20) for 8 weeks. Probiotic capsules consisted of strains of *Lactobacillus acidophilus*, *Lactobacillus casei* and *Bifidobacterium bifidum*. After 8 weeks of intervention, they showed that patients who received probiotic supplements had a significantly decreased Beck depression index (BDI) total score compared with the placebo [71]. Kazemi et al. (2019) conducted a study on 110 patients with mild and moderate depression who were allocated into three groups: probiotic group (*n* = 38), prebiotic group (*n* = 36) and placebo group (*n* = 36). After 8 weeks of intervention, they showed that the probiotic group had a significantly decreased BDI total score compared with the prebiotic and placebo groups [72]. Another study by Majeed et al. (2018) observed the effects of probiotics in 40 patients diagnosed with MDD and IBS. They were allocated into two groups: one group that received *Bacillus coagulans* MTCC 5856 and one group that received placebo. Changes in clinical symptoms were evaluated through questionnaires and they showed that there was a significant improvement in the *Bacillus coagulans* MTCC 5856 group compared with the placebo group, in both depression and IBS symptoms [73].

### 3.2. The Effects of Probiotics in Anxiety

Eskandarzadeh et al., (2021) conducted a study on the effects of probiotics as an adjuvant therapy in patients with GAD. The patients were allocated into two groups: one that received probiotics in addition to sertraline and one that received only sertraline. Probiotic capsules contained *Bifidobacterium longum*, *Bifidobacterium bifidum*, *Bifidobacterium lactis* and *Lactobacillus acidophilus*. After 8 weeks of treatment, they observed that the group that was given probiotic and sertraline had a significant decrease in the HAM-A score compared with the group that had only sertraline. There was also a decrease in the Beck anxiety inventory and QoL scores, but these changes showed no significant differences between the two groups [74].

### 3.3. The Effects of Probiotics in Schizophrenia

For schizophrenia, there were three randomized placebo-controlled double-blind trials and all of these belonged to the same research group. All three studies included patients diagnosed with schizophrenia or schizoaffective disorder and tested the same probiotic capsules containing *Lactobacillus rhamnosus* strain GG and *Bifidobacterium animalis subsp. lactis* strain Bb12 for a duration of 14 weeks. Psychiatric symptoms were assessed using the positive and negative syndrome scale (PANSS) and all three studies concluded that there was no significant difference on PANSS scores after supplementation with probiotics [75,76,77]. However, one of the studies showed that probiotics may have a beneficial role on adverse effects associated with antipsychotic use, such as weight gain and constipation [75]. Another study showed that probiotics have led to an increase in brain-derived neurotrophic factor (BDNF) levels [76]. A variation of BDNF levels has been associated with the common cognitive deficits found in chronic schizophrenia [78].

### 3.4. The Effects of Probiotics in Autism

Parracho et al., (2010) conducted a study on children diagnosed with ASD who were allocated into two groups: a group that received *Lactobacillus plantarum* and a placebo group. They showed that there were no significant differences between the two groups, neither in total behavior problem scores nor in terms of gastrointestinal symptoms associated with ASD [79]. Another study observed the effects of *Bifidobacterium infantis* and bovine colostrum product (BCP) combined and the effects of BCP alone in children with ASD and gastrointestinal problems. They showed that children who received only BCP had a significant reduction in some behaviors such as irritability and hyperactivity according to the aberrant behavior checklist questionnaire, while children who received both probiotic and BCP had a significant reduction in lethargy. Regarding adaptive and repetitive behaviors, there was no difference between the two groups. However, this study did not have a control group for comparison and included a small number of patients (*n* = 8) [80].

## 4. The Role of Postbiotics in Psychiatric Disorders

Postbiotics are inactivated microbial cells, with or without metabolites or cell components, that have health benefits through enhancement of epithelial barrier function, modulation of host-microbiota and immune responses and signaling via the nervous system [81]. There are recent studies showing the beneficial effects of postbiotics in depression [82,83,84]. Maehata et al., (2019) investigated the antidepressant effects of heat-killed *Lactobacillus helveticus* strain MCC1848 in a mouse model of subchronic and mild social defeat stress; the mice showed a significantly increased interaction time in the social interaction test which indicates anxiolytic or antidepressant-like effects [82]. Warda et al., (2019) investigated the effects of the postbiotic ADR-159 on mice. ADR-159 contains a co-fermentate of heat-killed *Lactobacillus fermentum* and *Lactobacillus delbrueckii*. They showed that the administration of the postbiotic for 3 weeks changed the composition of the microbiota, decreased locomotor activity in the open-field test and reduced the baseline corticosterone levels, which may indicate a potential action on the HPA axis [83]. The effects of postbiotics in humans were investigated in a study on 60 young adult students preparing for the national examination for medical practitioners. This group served as a model of chronic psychological stress. They were given heat-inactivated washed *Lactobacillus gasseri* CP2305, which significantly reduced anxiety and sleep disturbance compared with the placebo group. Furthermore, analysis of the gut microbiome showed a decrease of *Bifidobacterium* spp. and an increase of *Streptococcus* spp. [84].

## 5. Conclusions

This review assessed the changes in the gut microbiome that occurred within four psychiatric diseases (anxiety, depression, autism and schizophrenia), both in humans and in animal models. In addition, the impact of the use of probiotics and postbiotics in these pathologies was analyzed.

There seems to be a connection between the gut microbiome and certain psychiatric pathologies. This supposition is valid for anxiety, as well as for depression, schizophrenia and autism. In anxiety, eight bacterial taxa exhibited a lower prevalence compared with HCs: *Bacteroides*, *Faecalibacterium*, *Eubacterium*, *Lachnospira*, *Butyricicoccus*, *Sutterella*, *Bacteroidetes*, *Ruminococcus gnavus* and *Fusobacterium.* In both anxiety and depression patients, there was a lower prevalence of *Ruminococcus* compared with the control group. The presented studies included patients only with anxiety or depression but not with both conditions. In MDD patients, there seemed to be a higher abundance of *Alistipes*, *Bacteroides*, *Parabacteroides*, *Roseburia*, *Oscillibacter*, *Clostridium*, *Blautia*, and *Phascolarctobacterium* and a lower abundance of *Escherichia/Shigella*, *Dialister*, *Faecalibacterium*, *Prevotella* and *Ruminococcus* compared with HCs. In ASD patients, there was a higher abundance of *Bacteroides*, *Parabacteroides*, *Lactobacillus*, *Desulfovibrio*, *Sutterella* and *Clostridium* and reduced abundances of *Bifidobacterium*, *Dialister*, *Faecalibacterium*, *Streptococcus*, *Phasolarctobacterium*, *Veillonella*, *Prevotella* and *Coprococcus* compared with HCs. In SCZ, there was an increase in *Veillonellaceae*, *Prevotellaceae*, *Bacteroidaceae* and *Coriobacteriaceae* and a decrease in *Lachnospiraceae*, *Ruminococcaceae* and *Enterobacteriaceae* compared with HCs.

There is still a lack of studies on humans in this area. Future research is needed to properly demonstrate the gut microbiome’s part in mental health, how gut microorganisms could impact brain function and to possibly establish treatments for psychiatric pathologies that would directly target the microbiome.

Studies on the effects of probiotics in psychiatric disorders showed that probiotics have a beneficial impact in reducing the severity of the symptoms associated with depression and anxiety but not on those associated with SCZ or ASD. Some studies showed that they also have a role in reducing the gastrointestinal symptoms associated with antipsychotic treatment. However, there is a lack of clinical trials regarding the effects of probiotics in the psychiatric field; thus, future research is needed to establish the potential of probiotics as an adjuvant therapy in different psychiatric disorders.

## Figures and Tables

**Figure 1 microorganisms-10-02436-f001:**
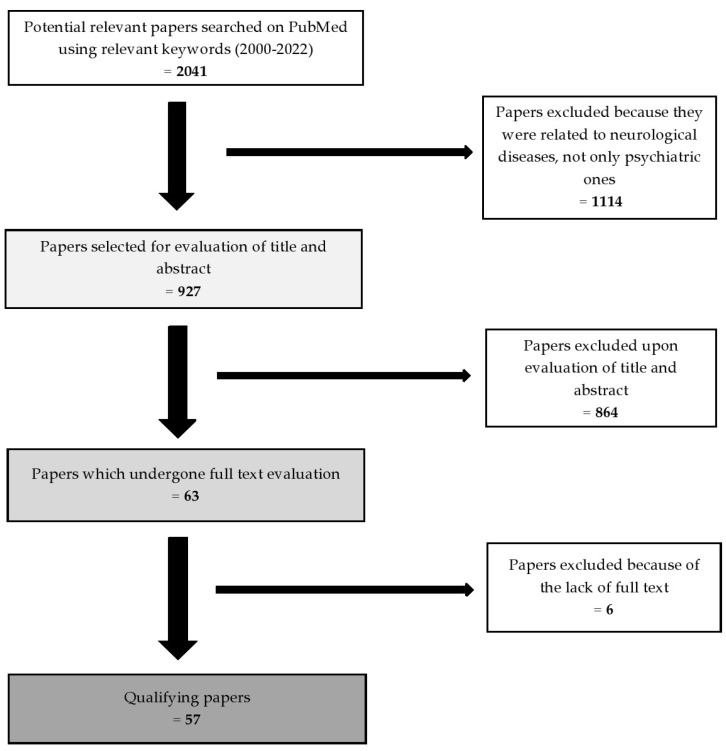
The selection process of the articles.

**Table 1 microorganisms-10-02436-t001:** Variations of gut microbiome in anxiety.

Disorder	Subjects	Test	Increased levels	Decreased levels	Study
Anxiety	Mice	Gut microbiota analysis	*Ruminiclostridium*, *Ruminococcaceae_UCG-014 Oscillibacter* *Clostridiales* *Clostridiaceae*	*Bacteroides* *Bacteroidaceae*	Jin et al., (2021) [27]
Anxiety	Mice	Gut microbiota analysis	*Ruminococcaceae*	-	Kang et al., (2014) [28]
Anxiety	Mice	Gut microbiota analysis	*Alistipes*	-	Bangsgaard Bendtsen et al., (2012) [29]
Anxiety	Mice	Gut microbiota analysis	*Citrobacter rodentium*	-	Lyte et al., (2006) [30]
GAD	40 patients, age 21–55 36 HCs, age 21–55	Fecal microbiota analysis	-	*Faecalibacterium* *Eubacterium* *Lachnospira* *Butyricicoccus* *Sutterella* *Bacteroidetes* *Ruminococcus gnavus* *Fusobacteium*	Jiang et al., (2018) [32]

GAD—generalized anxiety disorder; HCs—healthy controls.

**Table 2 microorganisms-10-02436-t002:** Variations of gut microbiome in depression.

Disorder	Subjects	Test	Increased Levels	Decreased Levels	Study
MDD	28 rats 34 patients, mean age 45.8 33 HCs, mean age 45.8	FMT from patients to rats	At the family level, *Thermoanaerobacteriaceae;* at the genus level, *Eggerthella* *Holdemania* *Gelria* *Turicibacter* *Paraprevotella* *Anaerofilum*	At the family level, *Prevotellaceae;* at the genus level, *Prevotella* *Dialister*	Kelly et al., (2016) [40]
MDD	5 patients, male, age 27–61 5 HCs, male, age 29–62	Fecal microbiota analysis	*Actinobacteria*	*Bacteroidetes*	Zheng et al., (2016) [39]
Active-MDD	29 patients, age 18–40 30 HCs	Fecal microbiota analysis in comparison with HCs	*Acidaminococcaceae Enterobacteriaceae Fusobacteriaceae* *Porphyromonadaceae Rikenellaceae* *Alistipes* *Blautia* *Clostridium* XIX *Lachnospiracea incertae sedis Megamonas* *Parabacteroides* *Parasutterella Phascolarctobacterium* *Oscillibacter* *Roseburia*	*Bacteroidaceae* *Erysipelotrichaceae* *Lachnospiraceae Prevotellaceae* *Ruminococcaceae* *Veillonellaceae*	Jiang et al., (2015) [43]
Responding-MDD	17 patients, age 18–40 30 HCs	Fecal microbiota analysis in comparison with HCs	At the family level, *Acidaminococcaceae* *Bacteroidaceae* *Enterobacteriaceae Porphyromonadaceae* *Rikenellaceae;* at the genus level, *Alistipes* *Bacteroides* *Parabacteroides Phascolarctobacterium Roseburia*	At the family level, *Lachnospiraceae* *Ruminococcaceae* *Veillonellaceae;* at the genus level, *Escherichia/Shigella Oscillibacter* *Dialister* *Faecalibacterium* *Prevotella* *Ruminococcus*	Jiang et al., (2015) [43]
MDD	37 patients, mean age 42.9 18 HCs, mean age 46.1	Fecal microbiota analysis	At the order level, *Bacteroidales;* at the genus level *Oscillibacter* *Alistipes*	*Lachnospiraceae*	Naseribafrouei et al., (2014) [44]
MDD	Belgian Flemish Gut Flora Project (*n* = 1054)	Fecal microbiota analysis	*Bacteroides* enterotype 2	*Coprococcus* *Dialister*	Valles-Colomer et al., (2019) [42]

MDD—major depressive disorder; HCs—healthy controls.

**Table 3 microorganisms-10-02436-t003:** Variations of gut microbiome in autism.

Disorder	Subjects	Test	Increased Levels	Decreased Levels	Study
ASD	143 ASD children, mean age 4.937 +/− 0.155 143 TD individuals, mean age 5.189 +/− 0.170	Fecal microbiota analysis	*Firmicutes* *Proteobacteria* *Actinobacteria* *Dialister* *Escherichia-Shigella Bifidobacterium*	*Bacteroidetes*	Dan et al., (2020) [56]
ASD	78 ASD children, mean age 4.96 +/− 1.01 58 HCs, mean age 4.90 +/− 0.97	Fecal microbiota analysis	*Bacteroides* *Parabacteroides* *Sutterella* *Lachnospira* *Bacillus* *Bilophila* *Lactococcus* *Lachnobacterium* *Oscillospira*	*-*	Zhai et al., (2019) [57]
ASD	48 children, age 2–7 48 HCs, age 4	Fecal microbiota analysis	At the phylum level, *Bacteroidetes;* at the genus level, *Bacteroides* *Prevotella* *Lachnospiracea_incertae_sedis* *Megamonas*	At the phylum level, *Firmicutes Proteobacteria Verrucomicrobia;* at the genus level, *Clostridium XlVa Eisenbergiella Clostridium IV* *Flavonifractor* *Escherichia/Shigella* *Haemophilus* *Akkermansia* *Dialister*	Zou et al., (2020) [58]
ASD	33 ASD children, age 2–13 7 non-autistic siblings 8 HCs	Fecal microbiota analysis	*Desulfovibrio* *Bacteroidetes vulgatus*	-	Finegold et al., (2010) [46]
ASD	41 ASD children, age 2–18 45 non-autistic siblings 45 HCs	Fecal microbiota analysis	*Bacteroides* *Ruminococcus*	-	Ahmed et al., (2020) [59]
ASD	77 ASD children, age 2–7 50 HCs	Fecal microbiota analysis	unidentified *Lachnospiraceae* *Clostridiales* *Erysipelotrichaceae* *Dorea* *Collinsella* *Lachnoclostridium*	*Bacteroides* *Faecalibacterium* *Parasutterella Paraprevotella*	Ding et al., (2020) [60]
ASD	20 ASD children and 20 HCs, age 3–16	Fecal microbiota analysis	*Akkermansia*	*Prevotella* *Coprococcus* *Veillonellaceae*	Kang et al., (2013) [61]
ASD	23 ASD children 22 non-ASD siblings 9 HCs	Fecal microbiota analysis	*Sutterella* *Ruminococcus torques*	*-*	Wang et al., (2013) [62]

ASD—autism spectrum disorder; HCs—healthy controls.

## Data Availability

Data are contained within the article.

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
