# Peer review of "The Role of the Gut Microbiome in Psychiatric Disorders"

_microorganisms, 2022, doi:10.3390/microorganisms10122436_

Round 1
Reviewer 1 Report
Manuscript is well written. Suggestions and questions are given below:
-How did the authors choose the studies whose results they discussed (year of publication, high citation index, was this a random choice?)
- L 136 A fecal microbiota transplantation (FMT) to rats was conducted.
Is fecal transplantation for the treatment of psychiatric diseases only in an experimental testing phase or this procedure can be carried out in certain hospitals with patients?
-Line 118 Oral administration of Citrobacter rodentium and Campylobacter jejuni showed that such pathogenic bacteria can activate vagal pathways which produce a stress response [31]
Campylobacter jejuni is a food pathogen, given that this manuscript deals with the connection between nutrition and psychiatric diagnoses this should be emphasized.
-Line 160 in response to 4 weeks of treatment.
Treatment of what?
-Line 322 Conclusion
Depression and anxiety often occur together. What is the prevalence of Ruminococcus in patients diagnosed with depression and anxiety compared to the control group. Did presented studies excluded patients with both conditions, i.e., did the included patients had only depression or anxiety and not both conditions at the same time (if this information is not stated in the studies, it should be highlighted in this manuscript)
-L 251 The effects of probiotics in psychiatric disorders
In addition to the influence of probiotics, is the influence of postbiotics (inactivated microbial cells or cellular components) on psychiatric diseases also known?
Reviewer 2 Report
This is a descriptive review about the gut-brain axis and the role of in modulating the gut-brain axis. The review is appropriate, useful, and timely review on nutritional (probiotics) and preventive behavioral medicine and psychiatry.
In my opinion, is optimal to include a representative figure/cartoon of the main review-topics.
Reviewer 3 Report
Reviewer comments and suggestions
In this review, the author highlighted the psychiatric disorders, namely anxiety, depression, autism, and schizophrenia based on the relevant studies on rodents, as well as human studies that connects with gut microbiome.
The results obtained in murine models align with the results acquired from patients, however fewer studies regarding anxiety were conducted on humans, and additionally the authors discuss the beneficial role of probiotics administered as an aid to the psychiatric medication.
The manuscript requires additional points to be incorporated in the manuscript based on the below preliminary comments.
- Line 20 Need to complete the points
- Line 24 please clear the sentence
- Line 37-38 needs a suitable reference
- In addition to reference 3, the authors need to add more relevant studies (for citing articles)
- Line 63 all references need to be discussed comprehensively
- After going through the manuscript, I suggest that the authors need to add one table and 2-3 figures discussing all important points on gut microbiome in psychiatric disorders
- It should also mention in the article what new topics, the authors included in the paper.
- All references should be modified based on the MDPI journals.
Round 2
Reviewer 2 Report
Dear authors,
the changes we made align with my considerations.
Reviewer 3 Report
All comments were incorporated in the manuscript. No more comments.